# A novel dental biosafety device to control the spread of potentially contaminated dispersion particles from dental ultrasonic tips

Victor Angelo Martins Montalli[1,2]☯*, Aguinaldo Silva Garcez[1]☯, Laís Viana Canuto de Oliveira[1]☯, Marcelo Sperandio[2]☯, Marcelo Henrique Napimoga[3]☯, Rogério Heládio Lopes Motta[4]☯

1 Division of Microbiology, Faculdade São Leopoldo Mandic, Campinas, São Paulo, Brazil, 2 Division of Oral Medicine, Faculdade São Leopoldo Mandic, Campinas, São Paulo, Brazil, 3 Division of Immunology, Faculdade São Leopoldo Mandic, Campinas, São Paulo, Brazil, 4 Division of Pharmacology, Faculdade São Leopoldo Mandic, Campinas, São Paulo, Brazil

☯ These authors contributed equally to this work.
* victor.montalli@slmandic.edu.br, victormontalli@gmail.com

**Data Availability Statement:** All relevant data are within the manuscript and its Supporting Information files.

## Abstract

Strategies to return to dental practice in pandemic times is a new challenge due to the generation and spread of potentially contaminated dispersion particles (PCDP) that may contain the SARS-CoV-2, the etiological factor of the COVID-19 disease. Due to the significant dispersion of PCDP in the dental environment, the use of equipment such as ultrasonic tips have been inadvisable during the pandemic. Several clinical procedures, however, benefit from the use of such equipment. Thus, using a microbial dispersion model of PCDP, the aim of this study was to compare the dispersion caused by the dental drill (DD) an ultrasonic tip (UT) alone and the UT coupled with a Spray control (SC) device. The DD, UT (with or without the SC) were activated for one minute having had the water from the reservoir replaced with a suspension of *Lactobacillus casei* Shirota ($1.5 \times 10^8$ CFU/mL). Petri dishes containing MRS agar were positioned at 50cm, 100cm and 150cm from the headrest of the dental chair at different angles (0 degree and 90 degrees). At 50 cm, the mean CFU (standard deviation) of *L. casei* Shirota was 13554.60 (4071.03) for the DD, 286.67 (73.99) for the US (97.89% reduction), and 4.5 (0.58) CFU for the UT-SC (p < 0.0001), establishing a further 98.43% reduction between UT and UT with SC. The UT with SC model proved effective in reducing dispersion from the UT, endorsing its use as an additional strategy to reduce PCDP in the dental environment in times of pandemic.

## Introduction

Since December 2019, the coronavirus disease 2019 (COVID-19), caused by the SARS-CoV-2 has rapidly spread worldwide [1]. SARS-CoV-2 is transmitted by contact with contaminated individuals with symptoms of cold and cough and the main route of transmission is the respiratory tract [2]. Coronaviruses including the SARS-CoV-2 have been shown to spread via aerosols and ventilation systems, inducing nosocomial infections as well as extensive hospital

**Funding:** The authors received no specific funding for this work.

**Competing interests:** The authors have declared that no competing interests exist.

outbreaks [3]. As evidence of such spread, an outbreak of COVID-19 was reported onboard of the U.S.S. Theodore Roosevelt, a nuclear-powered aircraft carrier, in which the close proximity conditions to both asymptomatic and pre symptomatic infected crew members drove the outbreak [4]. Because of that, COVID-19 was classified as a highly contagious disease with high morbidity and mortality [5].

Due to such high transmission, dental professionals face an unprecedented challenge in terms of providing primary dental care [6]. After a period in which dental practices were closed, reopening requires extra care in biosafety in order to prevent cross contamination. It has been demonstrated under experimentally induced aerosol conditions that the SARS-CoV-2 remained viable in aerosols for at least 3 h and on surfaces for up to 72 h. The longest viability was found on stainless steel (estimated mean half-life: 5.6 h) and plastic surfaces (estimated mean half-life: 6.8 h) [7]. Thus, dental associations worldwide have issued several recommendations for reopening, leading to important innovations in terms of personal protection equipment (PPE), such as individual dental biosafety barriers [8] to reduce oropharyngeal aerosol spread in closed environments [9, 10].

In dental practice, various dental procedures such as ultrasonic scaling, crown preparation, caries excavation, etc. can generate and release droplets of saliva, blood and other particles into the air, which may contain potentially infectious blood borne and airborne pathogens [11]. Some of the most intensive aerosol and splatter come from ultrasonic scalers [12].

The American Dental Association has warned that SARS-CoV-2 can also be spread in the dental environment by aerosols produced by air / water syringes, high and low speed handpieces, and when using ultrasonic scalers. Based on this context, the Centers for Diseases Control (CDC) has recommended that professionals avoid the procedures that produce aerosol, as well as the use of dental handpieces and the air / water syringe whenever possible, thus prioritizing minimally invasive restorative procedures [13].

Despite such recommendation during the pandemic, the use of ultrasonic tips has brought important advances in dentistry, such as pulp chamber access; core removal, cavity preparations, surgery, removal of residual restoration material, osteotomy etc. These in turn have translated into improved general health for patients and optimized the work of professionals caring for such individuals [14, 15]. Ultrasonic tips are, however, based on ultrasonic vibration in the range of 25,000 to 32,000 cycles / sec [16], leading to an enormous spread of potentially contaminated dispersion particles (PCDP), hence the need for methods to reduce PCDP formation.

Based on the aforementioned arguments, the present study aimed to evaluate a newly developed product known as 'spray control system' aimed at reducing PCDP from ultrasonic tips.

## Materials and methods

### Microorganism used

In summary, bacterial suspensions containing the microorganism *Lactobacillus casei* Shirota (Yakult Brasil Ltda, Lot # 0758) were used in the experiments. This strain was chosen because it is a bacterial species that poses no risk of environmental contamination and measures 0.5 μm (the SARS-CoV-2 virus measures 0.1 μm). Additionally, this microorganism has already been tested and validated for the dispersion model in a dental clinic environment in a previous study [8]. Thus, a viability test was performed to determine the initial concentration of $1.5x10^8$ CFU / mL of *L. casei* Shirota, based on the study by Marthi et al. [17].

### Generation of PCDP

PCDP generation in a dental clinic environment was simulated under two different conditions: 1) Activation of a dental drill at high speed and water cooling for 1 minute, 2) Activation

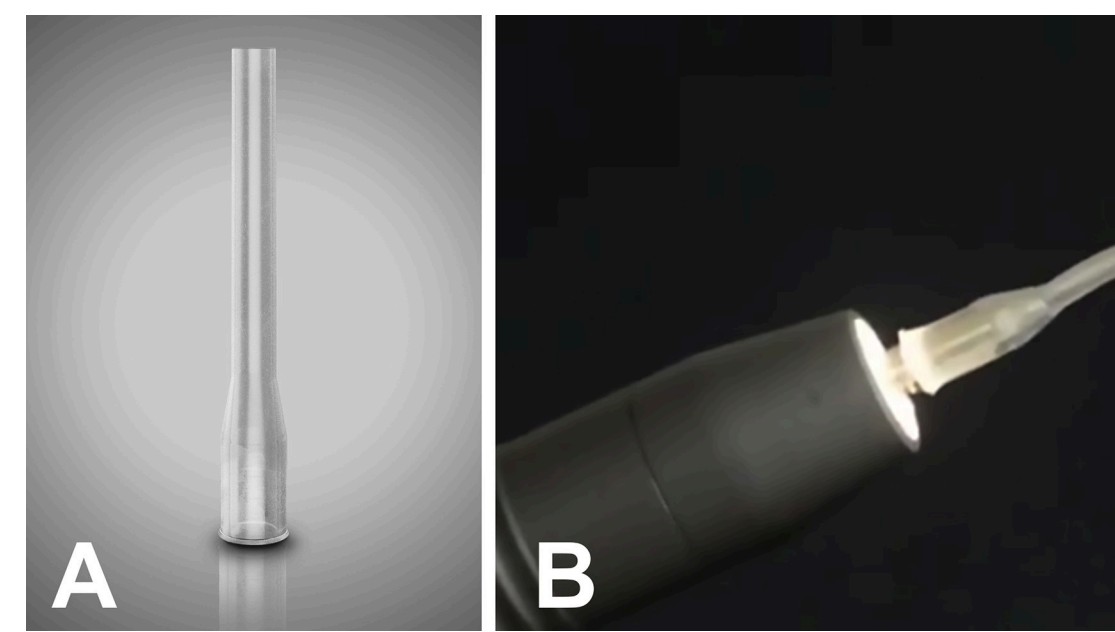

**Fig 1. Spray control system.** (A) freely or combined with ultrasonic tip (B).

of an ultrasonic tip for one minute; 3) Activation of an ultrasonic tip coupled with the spray control device. The generation of PCDP from ultrasonic instruments, both freely or combined with the spray control system, is illustrated in Fig 1.

For each condition tested, a suspension of *Lactobacillus casei* Shirota was added at a concentration of $1.5 \times 10^8$ CFU / mL in the water reservoir of a dental equipment. For the proposed model, a high-speed handpiece (Model 605C - Kavo, Brazil) was used. For the ultrasonic groups (UT and UT-SC), the equipment used was the Clinical Plus (CVDentus®, São José do Campos, Brazil). In both conditions, the equipment was activated for one-minute to simulate a clinical dental procedure.

## Spray control

The spray control device is made from autoclavable silicone with a flexible tubular shaft that fits over the ultrasonic tip to minimize spray, measuring 31.00 mm in length (15.70 mm of shaft) and 2.1mm in diameter. The length was based on the ultrasonic resonance points from the shaft to the active tip, i.e., such length ensured that the coolant was directed to the tip and that the active part in contact with the tooth would not be covered by the silicone (Fig 1). The sterile spray control device was placed over the tip of the ultrasonic device, which was subsequently activated as previously described.

## Thermal performance of ultrasonic tip

The thermal performance of the ultrasonic tip was investigated using images from a FLIR camera (Model: FLIR-E49001). This camera was used to capture thermal images from the ultrasonic tip with and without the spray control device and record temperature. An acrylic study model and an ultrasonic tip (model T0T, CVDentus®, São José do Campos, Brazil) were used. Thermal tests were performed using the ultrasonic tip (with and without the spray control device) in constant contact with the tooth at 30% and 100% power and at medium and

maximum cooling settings (Clinical Plus (CVDentus®, São José do Campos, Brazil) for 60 seconds.

### Microbiological and PCDP dispersion testing in a clinical setting

The dental clinic (12 m x 6.85 m x 2.5 m) where the study was conducted comprised 12 dental chairs (Dabi Atlante®, Brazil) 2 meters from each other. The dental clinic used for this study was closed to the public during the experiment, i.e. no patients were present, all doors and windows were kept shut to prevent air draft and the air conditioning system was off throughout the experiment.

The headrest of the dental chair was used as a reference point and one Petri dish (Ø 90mm x 15mm) containing enriched medium to Lactobacillus spp. (Lactobacilli MRS Agar, Neogen, Lot: 109503B) was positioned at 50 cm, 100 cm, and 150 cm at 0 degree and 90 degrees in relation to the headrest (Fig 2 Panels A and B). This experiment was then repeated on two separate occasions (i.e., triplicate), allowing a minimum of 24h between experiments. As negative controls, Petri dishes containing MRS medium were left open for 15 minutes prior to the test. Both the high-speed handpiece and the ultrasonic tip (UT and UT-SC) were activated over an acrylic tooth mounted in acrylic resin (Fig 2 Panels C–H). The high-speed turbine cooling volume flow was measured at 70mL / min. When testing the UT and UT-SC, the Clinical Plus equipment (CVDentus®, São José do Campos, Brazil) was set to the maximum coolant flow, which corresponds to 40 mL / min. With the handpieces activated using fluorescence Qscan (AioBio, Seoul, South Korea), images from PCDP were recorded (Fig 2 Panels F—H).

The Petri dishes were open immediately before activating the handpiece and were kept open for 15 minutes thereafter. They were then placed in an aerobic incubator (Tecnal, TE-399, Piracicaba, Brazil) for 48 hours. Colony Forming Units (CFU) were counted and Gram staining was performed to confirm the Lactobacilli culture. The tests were performed in triplicate. The size of the Petri dishes was 90 mm in diameter and the area 63.62 cm$^2$. Petri dishes who contained less than 300 CFU of *Lactobacillus casei* Shirota were counted in full. Petri dishes containing myriads of CFU had colonies counted based on three areas measuring 1 cm$^2$ each [18, 19]. Then, the average was calculated and multiplied by 63.62 (total area of the Petri dish). CFUs were counted manually (aided by a CFU counter) by a microbiology technician with over 10 years of experience.

### Statistical analysis

Data from both experiments were examined for normality by the Shapiro-Wilk test. As data demonstrated normality, all analyses were then performed using parametric methods. The differences in CFU for the different distances (50 cm, 100 cm and 150 cm) and groups (DD vs UT and UT vs UT-SC) were compared using One-Way ANOVA, followed by the Tukey test. Significance was established at 5%. All statistical analyses were performed on GraphPad Prism v6.0.

## Results

The spray control device reduced the temperature from 30.6 degrees Celsius (or 87.08 degrees F) to 28.5 degrees C (or 83.3 degrees F) at 30% power and medium irrigation. At maximum power (100%) and maximum irrigation, the temperatures recorded in the absence of the spray control device was 32.8 degrees C (91.04 degrees F) and 31.2 degrees C (88.16 degrees F) with the sleeve. Such findings are illustrated in Fig 3.

Regarding CFU counts and PCDP dispersion (when distance from the source was taken into account), an analysis of variance showed a significant difference was observed between

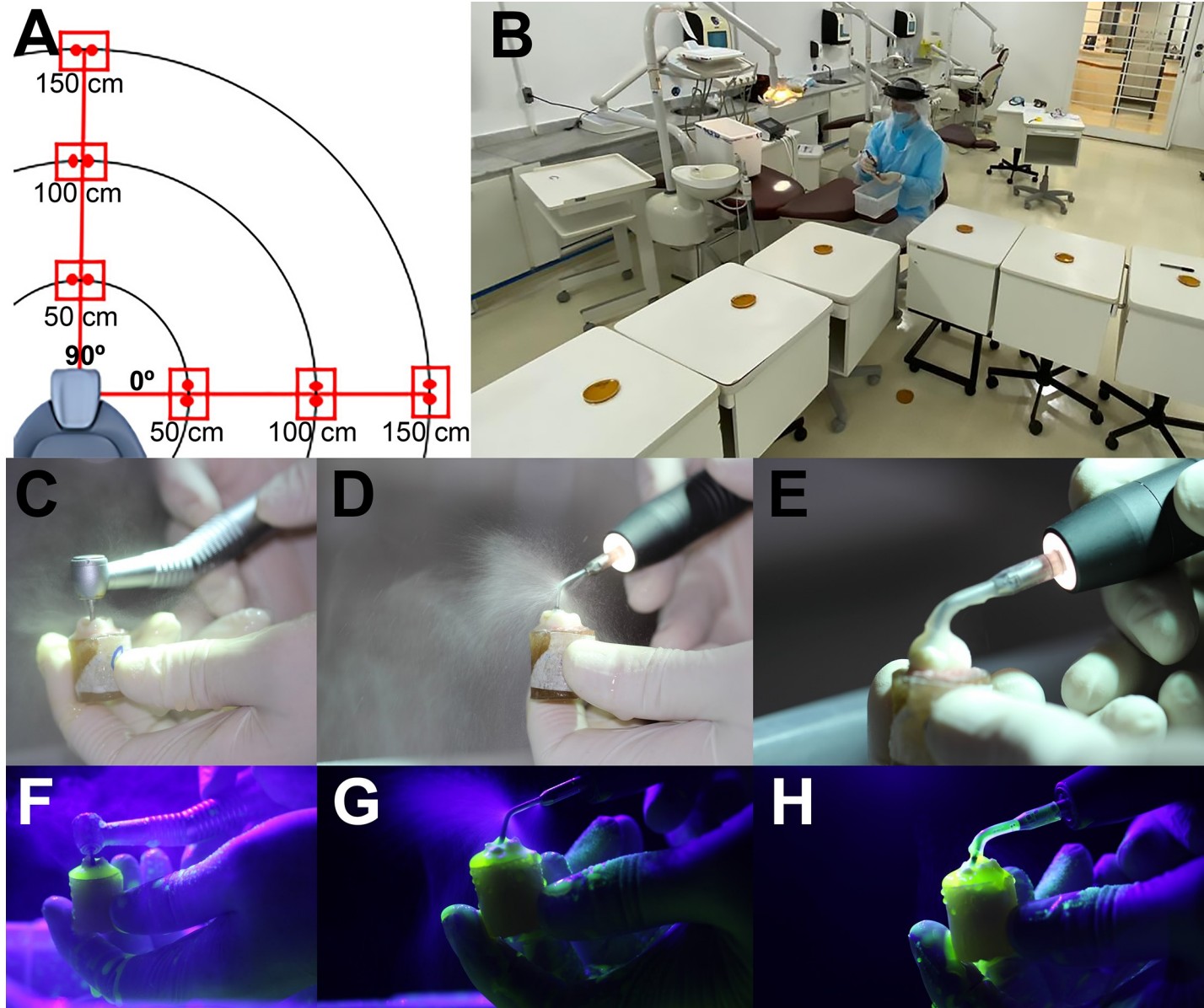

**Fig 2.** A) Schematic distribution of the petri dishes in relation to the headrest of the dental chair. B) Petri dishes opened for 15 minutes. Panels C and F show detail of the DD; Panels D and G show UT and Panels E and H show UT-SC generating droplets and aerosol. Panels F, G and H were used the image filter used for the accompanying figures at the bottom was the Qscan (AioBio, Seoul, South Korea).

the dental drill (DD) and the ultrasonic tip (UT), $F(5.30) = 703.90$, p = <0.0001, as well as the ultrasonic tip combined with the spray control device (UT-SC), $F(5.30) = 62.76$, p = <0.0001. The representative images of the Petri dishes of each group are found as (S1 Fig).

Regarding the dental drill (DD) at 90˚ and 0˚ angles, higher CFU counts were observed when compared with the ultrasonic tip (UT). Post hoc analyses using the Tukey's multiple comparisons criterion for significance indicated that at 50 cm, the mean and standard deviation (SD) of CFU of *Lactobacillus casei* Shirota for DD was 13554.60 (4071.03) CFU, while for UT was 286.67 (73.99) CFU (p <0.0001). At 100 cm, the mean DD was 7761.64 (2073.74) CFU and UT was 290.84 (96.21) CFU (p <0.0001). The mean (SD) at 150 cm for DD was 4464.00

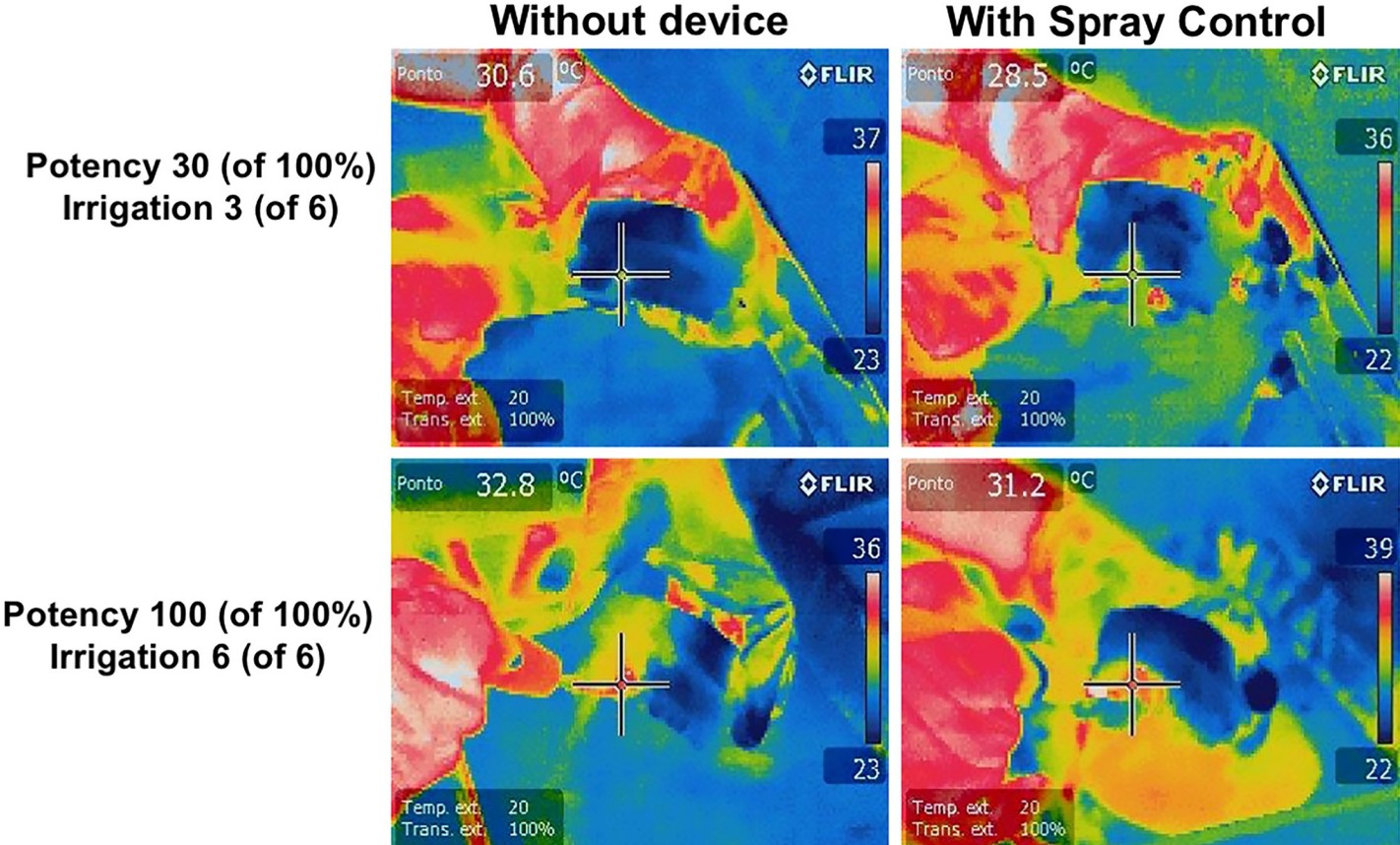

**Fig 3. Thermal imaging with and without the spray control device in place.**

(1011.65) CFU, while for UT was 344.33 (36.19) CFU (p <0.0001) (Fig 4). The mean difference between DD and UT was 96.42%. When using the spray control device on the ultrasound tip (UT-SC), the mean CFU counts (standard deviation) at 50 cm, 100 cm and 150 cm were 4.50 (0.58) CFU, 3.67 (1.06) CFU and 4.17 (3.47) CFU, which compared to the UT data translated into a significant difference (p <0.0001). The average difference between UT and UT-SC was 98.66%. When comparing DD and UT-SC, the mean difference was 99.95% (Fig 4).

## Discussion

The COVID-19 pandemic has been a great challenge for healthcare workers. The proportion of infected healthcare workers in April 2020 among confirmed cases was reported at 10% in Italy and 20% in Spain [20]. In the United States (US), approximately 3% of confirmed cases are healthcare workers, and 55% of these reported exposure to COVID-19 patients only in healthcare settings [21].

Due to the suspension of elective care during the pandemic in some countries and tighter rules on biosafety for dental procedures, some studies have shown low numbers of dental professionals diagnosed with COVID-19. In Brazil, the national report of the Ministry of Health showed that by August 2020, a total of 5,192 dental surgeons had been diagnosed with COVID-19, which corresponds to 0.17% of the total confirmed cases. Of the 241 deaths of health professionals recorded between March and August in Brazil, 13 cases were from dental professionals [22]. Data from The National Federation of Medical Doctors and Dentists (Italy)

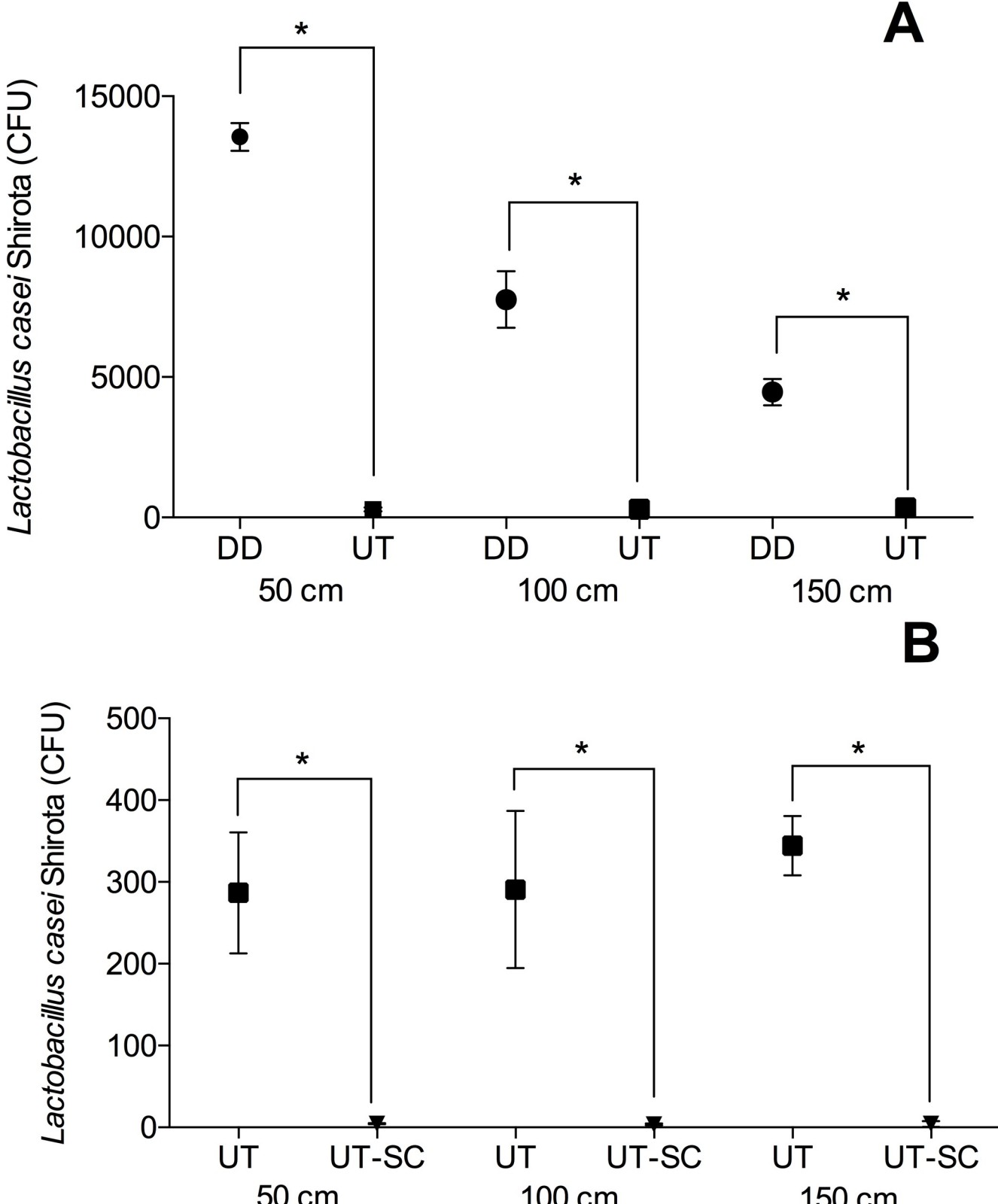

**Fig 4.** Mean and standard deviation of the CFU counts (*L. casei* Shirota) from the high-speed handpiece (DD), ultrasonic tip (UT) (A); and the ultrasonic tip combined or not with the spray control device (UT-SC) (B).* = p<0.0001. One-Way ANOVA, followed by the Tukey test.

updated on March 28[th] report the death of 86 medical doctors and 8 dentists over 92.472 confirmed cases [23]. This reinforces the concept that close contact with positive patients, whether symptomatic or not, exposes health workers to a higher risk of infection [24].

A previous study reported that the cell receptor for SARS-CoV-2 infection is by the angiotensin II-converting enzyme (ACE2), which is highly expressed in the oral mucosa. Notably, this receptor is present in large quantities in the epithelial cells of the tongue [25], indicating that the oral cavity could be a potential reservoir and hence a high-risk contact site for infection with this virus.

Considering this scenario, recommendations have been given to dental professionals to establish preventative screening strategies, such as to measure and record the temperature of each team and patient as a routine procedure. The pre-verification team should ask patients questions about their health status and contact or travel history [1]. However, further studies are still needed regarding the use of strategies that can minimize the generation and emission of PCDP. In this context, the present study evaluated different simulations of dental clinical procedures with a highly contaminated particles model using a high-speed dental drill (widely used in different dental procedures), the ultrasonic equipment, as well as the effectiveness of using a novel device coupled with the US to reduce PCDP.

The oral cavity is the second most diverse microbial environment in the human body and is home to more than 700 species of bacteria that colonize the soft tissues of the oral mucosa and the surfaces of teeth [26]. Although the methodology used was established with only one bacterial species (*Lactobacillus casei* Shirota), its main objective was to use a viable biological marker in the water reservoir of the dental chair, and consequently, provide a simulation of PCDP in dental procedures. This proposed model has already been used in a previous study and has shown no significant environmental risks [8], which is extremely important in this pandemic period. In addition, an enriched culture medium was used for the growth of *Lactobacillus*, thus providing more reliable results on PCDP generated during the experiments than germ-free models. A wide variety of sampling techniques and methods are found in the literature to evaluate contamination within the dental work environment using ultrasonic scalers [27]. Most such studies have, nonetheless, a common denominator, namely total CFU produced during different dental clinical procedures. According to Harrel & Molinari [28], this method provides a reasonable overall image of the increase in total bacterial CFU in the air of a specific procedure, though it does not differentiate between possible pathogenic species. It is important to highlight, however, that different dental procedures and settings may affect the dynamics of droplets and aerosol [29], for instance the fact that the experiment presented herein were performed with no mechanical ventilation. Besides, settle plates (such as Petri dishes) can only detect droplets that have fallen onto a surface, hence inadequate to assess variables such as air turbulence caused by movement.

As shown in the findings from the present study, the most important disperser of microorganisms was the high-speed DD. When compared to UT, a significant reduction in CFU count was observed in the order of 97%. When combined with the spray control device, the mean CFU count was reduced by 98% when compared to the uncovered UT. These data are important because different studies have highlighted the risk relating to dental procedures, particularly when high speed and ultrasonic instruments are used, where minimizing the generation of PCDP is desirable [30, 31]. A recent systematic review investigated contamination associated with routine dental procedures and concluded that ultrasonic scaling was one of the highest in contamination levels [29].

The present study has some limitations. The activation time of the dental equipment for one minute is not directly comparable to a standard dental procedure in terms of procedure or time, as it may vary considerably. Factors such as instrumentation time, sample exposure time

and sampling distance in relation to the working field can influence contamination spread [32, 33]. Furthermore, a realistic phantom head was not used (which could have provided an additional barrier to the spread of PCDP), and the ultrasonic tip was used only on the occlusal surface of the tooth, thus not necessarily representative of the potential pathogens from the mouth that could be picked up by the water spray and thrusted to the air. Also, this experimental model used small bacteria, as opposed to a virion like SARS-CoV-2, that settled after 15 minutes, though the remaining particles suspended in the air were not investigated. Such suspended PCDP is likely to represent a high inhalation risk during dental procedures and should therefore be further investigated. The findings presented herein, however, could be an invaluable indicator of reduced overall contamination when using the spray control device, as this generated significantly lower CFU counts when compared to the standard approach.

Additionally, the findings obtained herein corroborate several previous studies, demonstrating that the use of ultrasonic devices impact the contamination of the dental workplace with the production of aerosol and splatter [34, 35]. The significant reduction of PCDP with the spray control device, however, in addition to its practicality and low cost, justify this additional resource in the dental armamentarium. It is also important to emphasize that ultrasonic devices allow for reduced chair time [15], which would further endorse the implementation of such strategies during pandemics. Further studies are necessary to clarify whether reducing the production of PCDP from UT using spray control devices in an infected individual would translate into significant reduction in transmission risk.

The findings presented herein demonstrated that the use of the Spray Control device over the ultrasonic tip tested significantly reduced PCDP when compared to the free ultrasonic tip, which in a "best practice" approach justifies its use as an additional tool to minimize PCDP within the dental environment.

## Supporting information

**S1 Fig. Representative images of the Petri dishes of each group.** Dental drill (DD) and the ultrasonic tip (UT), as well as the ultrasonic tip combined with the spray control device (UT-SC).
(TIF)

**S1 Video.**
(MP4)

## Acknowledgments

The authors would like to thank the fundamental help of the Microbiology laboratory technicians at Faculdade São Leopoldo Mandic, Gilca Saba and Thiago Almeida.

## Author Contributions

**Conceptualization:** Victor Angelo Martins Montalli.

**Data curation:** Victor Angelo Martins Montalli, Aguinaldo Silva Garcez, Laís Viana Canuto de Oliveira, Marcelo Sperandio, Marcelo Henrique Napimoga.

**Formal analysis:** Victor Angelo Martins Montalli, Marcelo Henrique Napimoga.

**Methodology:** Aguinaldo Silva Garcez, Laís Viana Canuto de Oliveira, Marcelo Henrique Napimoga, Rogério Heládio Lopes Motta.

**Validation:** Marcelo Sperandio, Rogério Heládio Lopes Motta.

**Visualization:** Rogério Heládio Lopes Motta.

**Writing – original draft:** Victor Angelo Martins Montalli, Rogério Heládio Lopes Motta.

**Writing – review & editing:** Marcelo Sperandio.

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
