## [Decision Letter · Decision Letter 0]

13 Oct 2020

PONE-D-20-28707

The use of ultrasonic scalers combined with a novel dental biosafety device to control the dispersion of bioaerosols in the dental environment: additional care during pandemics.

PLOS ONE

Dear Dr. Montalli,

Thank you for submitting your manuscript to PLOS ONE. After careful consideration, we feel that it has merit but does not fully meet PLOS ONE’s publication criteria as it currently stands. Therefore, we invite you to submit a revised version of the manuscript that addresses the points raised during the review process.

We look forward to receiving your revised manuscript.

Kind regards,

Ratilal Lalloo

Academic Editor

PLOS ONE

Journal Requirements:

Reviewers' comments:

Reviewer's Responses to Questions

**Comments to the Author**

1. Is the manuscript technically sound, and do the data support the conclusions?

Reviewer #1: Partly

Reviewer #2: No

Reviewer #3: Partly

2. Has the statistical analysis been performed appropriately and rigorously? 

Reviewer #1: Yes

Reviewer #2: No

Reviewer #3: I Don't Know

3. Have the authors made all data underlying the findings in their manuscript fully available?

Reviewer #1: No

Reviewer #2: Yes

Reviewer #3: Yes

4. Is the manuscript presented in an intelligible fashion and written in standard English?

Reviewer #1: No

Reviewer #2: Yes

Reviewer #3: Yes

5. Review Comments to the Author

Reviewer #1: This paper has many limitations.

The introduction should cite more current references and refer to contact, droplet and airborne transmission routes for SARS-CoV-2 transmission. The current wording is confusing and somewhat inaccurate.

The paragraph that commences on line 90 is confusing. Endodontic surgery, veneer preparations and implant placement are not urgent or emergency dental procedures. There is no discussion around the point that such elective procedures can be deferred with no major impact on oral health.

The study aim should be refined to state that it was to assess a spray control system to reduce aerosol formation – since there is already much literature on dispersion of droplets and aerosols from powered dental equipment.

Details of the “spray control” device are not given – its length, and diameter (especially in relation to the scaler tip). How was the correct length determined? Was the same length used in multiple experiments?

What stops the tubing falling off the scaler tip and creating a risk for inhalation?

It is not clear how the “spray control” device that is proposed would allow the normal operation of an ultrasonic scaler, e.g. subgingivally, and to what extent the sleeve impairs proper operation of the scaler and increases tip temperature to a dangerous level – this could be shown using simple thermal imaging with and without the sleeve in place. The authors need to undertake tests to show that cavitation is still occurring at and around the tip, e.g. high speed imaging.

Why was the dental drill included? - as this is superfluous to the study design and hypothesis. There is no relevance to comparing DD vs US.

How was the concentration of the inoculum determined?

What was the water flow rate used with the scaler? There should have been a series of experiments testing different water flow rates as this is a critical variable.

Was there a control for a petri dish left exposed for 15 mins prior to any dental procedures being done? Was their checking that colonies seen and counted were ONLY lactobacilli?

Were experiments done on days when the clinic was not operating normally (i.e. no ambient air contamination)?

The authors need to some information (e.g. reflected ceiling plan diagram) showing the airconditioning outlets and inlets in the test clinic in relation to the test area.

How many times was the whole experiment repeated? The methods refers to only 2 - “both experiments”. Were these on different days? The photo shows 2 plates at each site – this does NOT constitute 2 experiments.

How were CFU counted – in detail? (how large were the culture plates, were colonies counted manually or digitally using software, etc) Are the authors claiming that they can count over 13,000 colonies on one single agar plate? Were counts made multiple times from the same plate to check for measurement error ?

The data for CFU for the ultrasonic show an increase with distance, which is rather counter-intuitive. This needs to be explained. If true, then samples should have also been done at much greater distances.

A major limitation in this study is that the conditions were not realistic in any way – the tooth was on the bench rather than in a phantom head or a mouth, and no normal high velocity evacuation was being used as is the case when an ultrasonic scaler is in used. As well, the scaler tip was used on the occlusal surface of the tooth, which is NOT a normal clinical technique. None of these are discussed.

The legend for the figure does not explain what the bottom series of “violet” images are showing.

There are style errors in references 5 and 7 – incorrect use of capitals in titles of articles

The operator shown working in the photo does not appear to be wearing their mask properly

Reviewer #2: The authors tried to explore how to reduce bacterial CFU formation with different dental equipment, which is of great significance to the field. However, the reviewer has the following concerns:

1. The title is too long for readers to attract attention. The authors should shorten the title to conclude their research.

2. The authors claimed their CFU formation is from bioaerosols, without any solid evidence. Probably they are only droplets and spatters in a shorter distance. Thus, the authors should be careful when claiming they are bioaerosols formation.

3. Since bacterial colonies are too subjective,

4. The authors should include their representative images in the result section. If there are more than 10,000 colonies, how did the authors count them if they are overlapped in the agar plate?

Reviewer #3: Thank you for the opportunity to review this interesting paper. The research question posed is certainly important during the present pandemic and more widely. I do however have a number of concerns which need to be addressed before I can recommend publication.

Introduction

There are several statements which should be supported by references which are not. For example: Line 77-80: “various dental procedures … can generate and release saliva particles, blood…”; Line 80-81: “Some of the most intensive aerosol and splatter come from ultrasonic scalers”; and statements in lines 83-89 relating to the American Dental Association and the CDC.

Materials and Methods

1. You should describe more fully how the L. casei Shirota suspension was made up; specifically, what was this prepared from (I note commercially available probiotic drink [Yakult] was diluted in a cited study), how was this diluted and with what, how was this stored, how long was it kept before use?

2. You should report what exact procedure you did with the air-turbine and ultrasonic scaler (US). You report these was activated for 1 minute but was this drilling/ scaling the tooth, or just holding over the tooth – this is not clear from the methods.

3. The irrigation rate (mL/minute) of the air-turbine and US should be reported.

4. Product details for the spray control device should begiven if this is a proprietary product. An image in the methods section specifically of this device would be of benefit to the reader.

5. What were the room dimensions of the area in which procedures were conducted, and what were the ventilation parameters (i.e. air exchanges per hour)? Was this a hospital clinic?

6. How many petri dishes were placed in each location, what size were they, and what was the total surface area of these at each site? It appears two were placed in each location from Figure 1, but this is unclear in the methods.

7. You report “The Petri dishes were kept open for 15 minutes” – was this 15 mins after the 1 min procedure (i.e. 16 mins total) or including the 1 min procedure (i.e. 15 mins total). How was a 15-minute exposure time decided upon?

8. Were any negative control conditions used, i.e. Petri dishes exposed without a procedure performed?

9. The legend for Figure 1 should come after the references. This legend reports: “The image filter used for the accompanying figures at the bottom was the Qscan (AioBio, Seoul, South Korea).” but this technique is not reported in the methods – it appears to be a form of direct fluorescence technique.

10. Did you conduct an a priori power calculation? If so, report this. ANOVA was used to test significance between distances, what test was used between experiments (i.e DD/US/US+Device)?

Results

1. You report that there was a reduction in CFU when using the US compared to the air-turbine; this is not a reduction – it is a difference. One would not use an US in place of an air-turbine (except in a select number of cases) and it is clearly a very different experimental condition to examine. Use difference instead of reduction throughout the manuscript.

2. Table 1: n is missing for the US-SC condition in the heading. No significance testing is reported – or at all elsewhere in the results for that matter. These need to be reported somewhere. Reporting of this data may be aided by also using a bar chart.

3. Table 2: again, no significance testing reported. It would be more appropriate to report the procedure with the lower CFU as a proportion of the procedure with the larger CFU, because as mentioned above you are not really showing a reduction, only a difference. For example, for DD versus US, it may be better to say that mean CFU for US was (x)% of that for DD rather than reporting a 95.57% reduction.

Discussion

1. In the first paragraph, you mention proportions of health care workers among confirmed COVID-19 cases in Italy and Spain – when was this? The time point is relevant in an evolving pandemic. Were these point prevalences at this particular time point or cumulative proportions of health workers who had been infected at any time?

2. In paragraph 2 you cite “The National Federation of Medical Doctors and Dentists”; which nation is this? You suggest that the data in this paragraph suggest that infection among oral health professionals is due to occupational exposure, but they do not unless you also know the corresponding rate of infection in the general population at that time to compare it to. You do not report this.

3. In line 209 you mention “record[ing] the temperature of each team and patient as a routine procedure”; what is the evidence for the effectiveness of this?

4. In line 227 you state that bacterial models of dental bioaerosol “thus [provide] more reliable results on dispersion of bioaerosols generated during the experiments than germ-free models”. Where is your evidence for this compared to other methodologies such as tracer dyes or particle counters? In fact, you need to discuss the evidence from such non-biological studies on dental aerosols and consider your findings in relation to these. A number of reviews of the literature in this area have recently been published and may be helpful in your locating of such primary sources of evidence: (https://www.scottishdental.org/wp-content/uploads/2020/08/Ventillation-Final-Copy-1.pdf ; https://www.sdcep.org.uk/published-guidance/covid-19-practice-recovery/rapid-review-of-agps/ ; https://www.medrxiv.org/content/10.1101/2020.08.28.20183475v1)

Limitations

You discuss the limitations of your study, however you have failed to identify the most significant limitations, as follows, which must be considered and reported:

1. You report that you have detected “aerosol” and have concluded that US produces less than the air-turbine. You have not identified the fact that your method will mainly detect droplet contamination, and the proportion of aerosols that settle onto Petri dishes in 15 minutes. You have detected neither aerosol which settles after 15 minutes, nor that which remains suspended in the air. This suspended aerosol is likely to be the greatest inhalation risk during a dental aerosol generating procedure. You should explore this limitation in the discussion and its implications to your findings.

2. The procedure you used is not equivalent to a standard dental procedure in terms of procedure or time. This will affect the ability to translate to actual dental procedures.

3. You examined to 1.5 m from the procedure, what happens beyond this? This limitation has relevance for open-plan dental clinics such as those where the investigation was performed.

4. You equate your findings with a bacterial biomarker to the carriage of virus (SARS-CoV-2). You need to discuss the limitation that a virion (which is much smaller, may be charged etc.) may not behave in the same way as a bacterium, and this will affect the translatability of your findings. The requirement for bacteria to be viable for you to detect it is also a limitation, and this may be different to the infectivity of virions at any given distance.

5. You do not discuss that placing the bacterial biomarker into dental irrigation systems will tell you where water spray will be distributed to, but not necessarily where potential pathogens from the mouth will be picked up by this water spray and distributed to. This affects how relevant your findings are to the risk of transmission from dental procedures in real-world conditions.

6. Finally, your findings show that US likely produces less contamination than an air-turbine, and that the spray control device reduces US contamination, but we still do not know what the risk threshold of SARS-CoV-2 is. For example, what viral load is required to cause COVID-19? Will the reduced amount of contamination from US with the spray device in an infected individual still be significant transmission risk? We do not yet know this and this should be discussed.

Again, thank you for an interesting and useful paper. I know that this may seem to be a detailed review, but your work needs to be better reported and interpreted before publication in my opinion. I would look forward to reading a revised manuscript if asked.

6. PLOS authors have the option to publish the peer review history of their article (what does this mean?). If published, this will include your full peer review and any attached files.

Reviewer #1: **Yes: **Laurence J. Walsh

Reviewer #2: No

Reviewer #3: **Yes: **James R Allison

---

## [Author Response · Author response to Decision Letter 0]

26 Nov 2020

Reviewer #1: This paper has many limitations.

1. The introduction should cite more current references and refer to contact, droplet and airborne transmission routes for SARS-CoV-2 transmission. The current wording is confusing and somewhat inaccurate.

A: Thank you for your suggestion. We have included additional information in the introduction to address the reviewer’s comment. Below the new information included in the revised manuscript. Please see new submitted version of the manuscript. 

“Coronaviruses including the SARS-CoV-2 have been shown to spread via aerosols and ventilation systems, inducing nosocomial infections as well as extensive hospital outbreaks [3]. As evidence of such spread, an outbreak of COVID-19 was reported onboard of the U.S.S. Theodore Roosevelt, a nuclear-powered aircraft carrier, in which the close proximity conditions to both asymptomatic and pre symptomatic infected crew members drove the outbreak [4].”

3. Bin SY, Heo JY, Song MS, Lee J, Kim EH, Park SJ, Kwon HI, Kim SM, Kim YI, Si YJ, Lee IW, Baek YH, Choi WS, Min J, Jeong HW, Choi YK. Environmental Contamination and Viral Shedding in MERS Patients During MERS-CoV Outbreak in South Korea. Clin Infect Dis. 2016 Mar 15;62(6):755-60. doi: 10.1093/cid/civ1020. Epub 2015 Dec 17. Erratum in: Clin Infect Dis. 2016 May 15;62(10):1328. Erratum in: Clin Infect Dis. 2016 Sep 15;63(6):851. PMID: 26679623; PMCID: PMC7108026.

4. Kasper MR, Geibe JR, Sears CL, Riegodedios AJ, Luse T, Von Thun AM, McGinnis MB, Olson N, Houskamp D, Fenequito R, Burgess TH, Armstrong AW, DeLong G, Hawkins RJ, Gillingham BL. An Outbreak of Covid-19 on an Aircraft Carrier. N Engl J Med. 2020 Nov 11. doi: 10.1056/NEJMoa2019375. Epub ahead of print. PMID: 33176077.

2. The paragraph that commences on line 90 is confusing. Endodontic surgery, veneer preparations and implant placement are not urgent or emergency dental procedures. There is no discussion around the point that such elective procedures can be deferred with no major impact on oral health.

A: Thank you for your careful reading. In fact, our first idea was to describe all possibilities to be using the ultrasonic tips, however some of the procedures described were not urgencies. In order to avoid misunderstanding we have rephrased the paragraph. Below the new information included in the revised manuscript. Please see new submitted version of the manuscript. 

“Despite such recommendation during the pandemic, the use of ultrasonic tips has brought important advances in dentistry, such as pulp chamber access; core removal, cavity preparations, surgery, removal of residual restoration material, osteotomy etc. These in turn have translated into improved general health for patients and optimized the work of professionals caring for such individuals [14,15]. Ultrasonic tips are, however, based on ultrasonic vibration in the range of 25,000 to 32,000 cycles / sec [16], leading to an enormous spread of droplets, hence the need for methods to reduce aerosol formation and prevent the dispersion of droplets”

3. The study aim should be refined to state that it was to assess a spray control system to reduce aerosol formation – since there is already much literature on dispersion of droplets and aerosols from powered dental equipment.

A: Thank you for your careful reading. We have rephrased the last paragraph of the introduction as requested. Below the new information included in the revised manuscript. 

“Based on the aforementioned arguments, the present study aimed to evaluate a newly developed product known as ‘spray control system’ aimed at reducing droplets and aerosols from ultrasonic tips.”

4. Details of the “spray control” device are not given – its length, and diameter (especially in relation to the scaler tip). How was the correct length determined? Was the same length used in multiple experiments?

A: Thank you for your careful reading. We have included additional information in the material and methods. Please, find below the new information added to the revised manuscript. 

“The spray control device is made from autoclavable silicone with a flexible tubular shaft that fits over the ultrasonic tip to minimize spray, measuring 31.00 mm in length (15.70 mm of shaft) and 2.1mm in diameter. The length was based on the ultrasonic resonance points from the shaft to the active tip, i.e., such length ensured that the coolant was directed to the tip and that the active part in contact with the tooth would not be covered by the silicone (Fig 1).”

 

Figure 1 - Spray Control system. (A) freely or combined with ultrasonic tip (B).

5. What stops the tubing falling off the scaler tip and creating a risk for inhalation?

A: The spray control system was developed for tips produced by the company CVDentus. The diameter of the silicone tube (2.1mm) is slightly narrower than the base of the ultrasonic tip (2.3mm), causing the device to lock firmly to the base of the tip. In addition, all ultrasonic tips manufactured by CVDentus have at least one bend along their shaft. Therefore, some manual force followed by gentle maneuvering are needed to free the spray control device from the ultrasonic tip.

The supplementary video demonstrates the appropriate handling and activation of the ultrasound tip.

6. It is not clear how the “spray control” device that is proposed would allow the normal operation of an ultrasonic scaler, e.g. subgingivally, and to what extent the sleeve impairs proper operation of the scaler and increases tip temperature to a dangerous level – this could be shown using simple thermal imaging with and without the sleeve in place. The authors need to undertake tests to show that cavitation is still occurring at and around the tip, e.g. high-speed imaging.

A: This was an important point raised by the referee. We have since performed additional experiments to investigate this and added the findings to the manuscript. The Spray Control sleeve ends before the active part of the tip begins. In the case of the tip studied, it allows 7mm of active instrument for contact with the target surface, thus not interfering with the normal performance of the equipment. Please find below the new data added to the revised manuscript. Please see new submitted version of the manuscript. 

Material and Methods

Thermal performance of ultrasonic tip 

The thermal performance of the ultrasonic tip was investigated using images from a FLIR camera (Model: FLIR-E49001). This camera was used to capture thermal images from the ultrasonic tip with and without the spray control device and record temperature. An acrylic study model and an ultrasonic tip (model T0T, CVDentus®, São José do Campos, Brazil) were used. Thermal tests were performed using the ultrasonic tip (with and without the spray control device) in constant contact with the tooth at 30% and 100% power and at medium and maximum cooling settings (Clinical Plus (CVDentus®, São José do Campos, Brazil) for 60 seconds. 

Results

The spray control device reduced the temperature from 30.6 degrees Celsius (or 87.08 degrees F) to 28.5 degrees C (or 83.3 degrees F) at 30% power and medium irrigation. At maximum power (100%) and maximum irrigation, the temperatures recorded in the absence of the spray control device was 32.8 degrees C (91.04 degrees F) and 31.2 degrees C (88.16 degrees F) with the sleeve. Such findings are illustrated in Fig 3.

Figure 3 - Thermal imaging with and without the spray control device in place.

7. Why was the dental drill included? - as this is superfluous to the study design and hypothesis. There is no relevance to comparing DD vs US.

A: The high-speed turbine was included as a positive control because it is known to generate great amounts of aerosol and, therefore, would serve as parameter for comparison and possibly encourage readers to consider replacing the turbine for ultrasonic tips whenever possible. Thus, the authors feel that this comparison of aerosol production between DD vs UT is highly relevant, which is why we decided to keep it.

8. How was the concentration of the inoculum determined?

A: This was based on the study by Marthi et al. (1990), Survival of bacteria during aerolization. According to the authors, bacterial survival during aerosolization is dependent on environmental conditions, e.g., temperature, relative humidity, droplet size, etc. So, in this study, the inoculum corresponded to a viable cell density of approximately 1.5x108 CFU per ml.

9. What was the water flow rate used with the scaler? There should have been a series of experiments testing different water flow rates as this is a critical variable.

A: The manufacturers themselves (CVDentus) recommend the maximum water flow (100%) for each tip and the equipment is fitted with and automation software for this purpose, in which vibration power is automatically set to the equivalent water flow. However, if the clinician wishes to manually set the water flow, it can be done at no detriment to the spray control sleeve as it was supposedly designed for the tips to work with the maximum coolant flow. In the tests, the water flow used was 100%, which is equivalent to 40 ml / min.

“When testing the UT and UT-SC, the Clinical Plus equipment (CVDentus®, São José do Campos, Brazil) was set to the maximum coolant flow, which corresponds to 40 mL / min.”

10. Was there a control for a petri dish left exposed for 15 mins prior to any dental procedures being done? Was their checking that colonies seen and counted were ONLY lactobacilli?

A: Thank you for your careful reading. We have added information to clarify that Gram Staining was performed. Please find below the added the new information. 

“As negative controls, Petri dishes enriched medium to Lactobacillus spp. were left open for 15 minutes prior to the test. Both the high-speed handpiece and the ultrasonic tip (UT and UT-SC) were activated over an acrylic tooth mounted in acrylic resin (Fig 2 Panels C – H)”. 

“With the handpieces activated using fluorescence Qscan (AioBio, Seoul, South Korea), images from droplets and aerosols were recorded (Fig 2 Panels F - H).

The Petri dishes were open immediately before activating the handpiece and were kept open for 15 minutes thereafter. They were then placed in an aerobic incubator (Tecnal, TE-399, Piracicaba, Brazil) for 48 hours. Colony Forming Units (CFU) were counted and Gram staining was performed to confirm the Lactobacilli culture. The tests were performed in triplicate.”

11. Were experiments done on days when the clinic was not operating normally (i.e. no ambient air contamination)?

A: The experiments were performed in May 2020 at the dental clinic of Faculdade São Leopoldo Mandic, during which period (from March 12th) all clinics and classes were interrupted following local government decrees to prevent the spread of COVID-19, whereby (elective) university facilities were accessible for in vitro research only. Therefore, all clinics, including the one used for the experiments reported on this manuscript, were closed, and no clinical procedures had been performed for > 50 days prior to experiments. To clarify this point in the study, we have added information to the Material and Methods. 

Please find below the new information included in the revised manuscript. 

“The dental clinic (12 m x 6.85 m x 2.5 m) where the study was conducted comprised 12 dental chairs (Dabi Atlante®, Brazil) 2 meters from each other. The dental clinic used for this study was closed to the public during the experiment, i.e. no patients were present, all doors and windows were kept shut to prevent air draft and the air conditioning system was off throughout the experiment.

The headrest of the dental chair was used as a reference point and one petri dish (Ø 90mm x 15mm) containing enriched medium to Lactobacillus spp. (Lactobacilli MRS Agar, Neogen, Lot: 109503B) was positioned at 50 cm, 100 cm, and 150 cm at 0 degree and 90 degrees in relation to the headrest (Fig 2 Panels A and B)”.

12. The authors need to some information (e.g. reflected ceiling plan diagram) showing the air-conditioning outlets and inlets in the test clinic in relation to the test area.

A: Further to the dental clinic being closed for 50 days +, as described previously, the text reads “all doors and windows were kept shut to prevent air draft and the air conditioning system was off throughout the experiment.” The authors hope that this information, combined with the aforementioned measurements of the clinical room where the experiments were performed will be satisfactory.

13. How many times was the whole experiment repeated? The methods refers to only 2 - “both experiments”. Were these on different days? The photo shows 2 plates at each site – this does NOT constitute 2 experiments.

A: Thank you for your careful reading. We have replaced the Fig 2B, as this was purely illustrative of a pilot set-up. The experiment was in fact performed once and repeated twice (i.e., 3 replicas) and for each experiment, a single plate was used at each distance point (50 cm, 100 cm and 150 cm). Additionally, each replica was performed at least 24h after the previous experiment. This information has been added to the methods to clarify this.

14. How were CFU counted – in detail? (how large were the culture plates, were colonies counted manually or digitally using software, etc) Are the authors claiming that they can count over 13,000 colonies on one single agar plate? Were counts made multiple times from the same plate to check for measurement error ?

A: “The size of the Petri dishes is 90 cm in diameter. Plate area = 63.62 cm2. Petri dishes containing less than 300 CFU of Lactobacillus casei Shirota were counted in full. Petri dishes containing myriads of CFU were counted in three representative areas (1 cm2 each) [18,19]. Then, an average was calculated and multiplied by 63.62 (total area of the Petri dish). CFUs were counted manually (aided by a CFU counter) by a microbiology technician with over 10 years of experience. This information has been added to the methods to clarify this.

18. Pasquarella C, Pitzurra O, Savino A. The index of microbial air contamination. J Hosp Infect. 2000 Dec;46(4):241-56. doi: 10.1053/jhin.2000.0820. PMID: 11170755.

19. Viani I, Colucci ME, Pergreffi M, Rossi D, Veronesi L, Bizzarro A, Capobianco E, Affanni P, Zoni R, Saccani E, Albertini R, Pasquarella C. Passive air sampling: the use of the index of microbial air contamination. Acta Biomed. 2020 Apr 10;91(3-S):92-105. 

15. The data for CFU for the ultrasonic show an increase with distance, which is rather counter-intuitive. This needs to be explained. If true, then samples should have also been done at much greater distances.

A: The slight increase observed in the distance of 150 vs 50 in the US group, for example, was not statistically significant. We established such distances based on a published preliminary study that describes the rationale for the distances investigated in the present study (reference 8). 

16. A major limitation in this study is that the conditions were not realistic in any way – the tooth was on the bench rather than in a phantom head or a mouth, and no normal high velocity evacuation was being used as is the case when an ultrasonic scaler is in used. As well, the scaler tip was used on the occlusal surface of the tooth, which is NOT a normal clinical technique. None of these are discussed.

A: The idea of the experiment was to observe the number of colony counts after the use of 3 tested groups (DD, UT, UT-SC) in a worst case scenario, where no high volume suction was available (regrettably, this is more realistic worldwide than one would have liked to accept) and no physical barrier (e.g., head, cheeks, lips) was present. All being said, the authors agree with the reviewer that such conditions are not representative of the ideal clinical set-up and could therefore consist in important elements of bias. We have therefore included this in the discussion of the revised manuscript, as follows: 

“Furthermore, a realistic phantom head was not used (which could have provided an additional barrier to the spread of droplets and aerosol), and the scaler tip was used only on the occlusal surface of the tooth, thus not necessarily representative of the potential pathogens from the mouth that could be picked up by the water spray and thrusted to the air. Also, this experimental model used small bacteria, as opposed to a virion like SARS-CoV-2, that settled after 15 minutes, though the remaining particles suspended in the air were not investigated. Such suspended aerosol is likely to represent a high inhalation risk during dental aerosol generating procedure and should therefore be further investigated.”

17. The legend for the figure does not explain what the bottom series of “violet” images are showing.

A: Thank you for your careful reading, we have included additional description to the legend. 

18. There are style errors in references 5 and 7 – incorrect use of capitals in titles of articles

A: Thank you. Modifications have been made to correct this. 

19. The operator shown working in the photo does not appear to be wearing their mask properly

A: Thank you for your attentive eye. We have replaced the photo in which the operator is wearing an N95 mask and face shield.

Reviewer #2: The authors tried to explore how to reduce bacterial CFU formation with different dental equipment, which is of great significance to the field. However, the reviewer has the following concerns:

1. The title is too long for readers to attract attention. The authors should shorten the title to conclude their research.

A: Thank you for your careful reading. We have shortened the title, as suggested. 

“A novel dental biosafety device to control the dispersion of bioaerosols from dental ultrasonic tips.”

2. The authors claimed their CFU formation is from bioaerosols, without any solid evidence. Probably they are only droplets and spatters in a shorter distance. Thus, the authors should be careful when claiming they are bioaerosols formation.

A: We used Lactobacillus casei Shirota as this microorganism is not usual in the clinical environment air. Thus, such microorganism was used as a mere surrogate marker of an infectious source (the point where the aerosol generator was activated) and we may therefore infer that the CFU counts came from the dental hand pieces.

3. Since bacterial colonies are too subjective, the authors should include their representative images in the result section. If there are more than 10,000 colonies, how did the authors count them if they are overlapped in the agar plate?

Thank you for your suggestion. We have included sample images from the plates of each group as a supplemental material. We also included additional information in the results section as reference to it. Please find below the new information included in the revised manuscript. 

“The size of the Petri dishes was 90 cm in diameter and the area 63.62 cm2. Petri dishes who contained less than 300 CFU of Lactobacillus casei Shirota were counted in full. Petri dishes containing myriads of CFU had colonies counted based on three areas measuring 1 cm2 each [18,19]. Then, the average was calculated and multiplied by 63.62 (total area of the Petri dish). CFUs were counted manually (aided by a CFU counter) by a microbiology technician with over 10 years of experience.”

Reviewer #3: Thank you for the opportunity to review this interesting paper. The research question posed is certainly important during the present pandemic and more widely. I do however have a number of concerns which need to be addressed before I can recommend publication.

Introduction

There are several statements which should be supported by references which are not. For example: Line 77-80: “various dental procedures … can generate and release saliva particles, blood…”; Line 80-81: “Some of the most intensive aerosol and splatter come from ultrasonic scalers”; and statements in lines 83-89 relating to the American Dental Association and the CDC.

A: Thank you for your suggestion. We have included additional information in the introduction to clarify this. A new reference has also been included in the revised manuscript. Please, see the revised version of the manuscript. 

11. Clementini M, Raspini M, Barbato L, Bernardelli F, Braga G, Di Gioia C, Littarru C, Oreglia F, Brambilla E, Iavicoli I, Pinchi V, Landi L, Marco Sforza N, Cavalcanti R, Crea A, Cairo F. Aerosol transmission for SARS-CoV-2 in the dental practice. A review by SIdP Covid-19 task-force. Oral Dis. 2020 Oct 29. doi: 10.1111/odi.13649. Epub ahead of print. PMID: 33124127.

Materials and Methods

1. You should describe more fully how the L. casei Shirota suspension was made up; specifically, what was this prepared from (I note commercially available probiotic drink [Yakult] was diluted in a cited study), how was this diluted and with what, how was this stored, how long was it kept before use? 

In preliminary tests, we identified that 1mL of Yakult bears 9.02x108 CFU of L. casei Shirota. The Yakult flask contains 78mL, which was diluted in 390mL of saline, therefore: 𝑓𝐶𝑓𝑓=1.50𝑓𝑥108 CFU per 𝑓𝑚𝑓𝐿. The suspension was prepared immediately before experimentation. This information has been added to the methods to clarify this. Thanks for bringing attention to this important methodological information.

2. You should report what exact procedure you did with the air-turbine and ultrasonic scaler (US). You report these was activated for 1 minute but was this drilling/ scaling the tooth, or just holding over the tooth – this is not clear from the methods.

A: Thank you for your careful reading. We have rephrased the sentence to prevent misunderstanding. Please find below the new information included in the revised manuscript. 

“Both the high-speed handpiece and the ultrasonic tip (UT and UT-SC) were activated over an acrylic tooth mounted in acrylic resin (Fig 2 Panels C – H)…”

3. The irrigation rate (mL/minute) of the air-turbine and US should be reported.

A: Reviewer 1 has also raised this point and new information has been added to the revised manuscript, as follows:

A: The manufacturers themselves (CVDentus) recommend the maximum water flow (100%) for each tip and the equipment is fitted with and automation software for this purpose, in which vibration power is automatically set to the equivalent water flow. However, if the clinician wishes to manually set the water flow, it can be done at no detriment to the spray control sleeve as it was supposedly designed for the tips to work with the maximum coolant flow. In the tests, the water flow used was 100%, which is equivalent to 40 ml / min. 

“The high-speed turbine cooling volume flow was measured at 70mL / min.”

4. Product details for the spray control device should begiven if this is a proprietary product. An image in the methods section specifically of this device would be of benefit to the reader.

A: Thank you for your suggestion. We have included additional information regarding the dimensions of the spray control as also suggested by the reviewer 1. Also, we have included a Figure to illustrate the device. Please see new figure 1 and description therein.

5. What were the room dimensions of the area in which procedures were conducted, and what were the ventilation parameters (i.e. air exchanges per hour)? Was this a hospital clinic?

A: Thank you for your careful reading. Faculdade São Leopoldo Mandic in Campinas, Brazil has 12 dental clinics fitted with 12 dental chairs each dedicated to post graduate students, though the building where the dental clinics are set up does not constitute a hospital, as far as local regulations are concerned. Each clinic has its own individual air conditioning system as well as windows. We have included the dimension of the clinic used for this test. As already described in the methodology (please see below), the doors and windows were kept shut to prevent air draft and the air conditioning equipment remained off throughout the experiment.

“The dental clinic (12 m x 6.85 m x 2.5 m) where the study was conducted comprised 12 dental chairs (Dabi Atlante®, Brazil) 2 meters from each other. The dental clinic used for this study was closed to the public during the experiment, i.e. no patients were present, all doors and windows were kept shut to prevent air draft and the air conditioning system was off throughout the experiment.”

6. How many petri dishes were placed in each location, what size were they, and what was the total surface area of these at each site? It appears two were placed in each location from Figure 1, but this is unclear in the methods.

A: Thank you for your careful reading. We have modified the text in the methods to clarify this. Please see below the information added to the revised manuscript.

The headrest of the dental chair was used as a reference point and one petri dish (Ø 90mm x 15mm) containing enriched medium to Lactobacillus spp. (Lactobacilli MRS Agar, Neogen, Lot: 109503B) was positioned at 50 cm, 100 cm, and 150 cm at 0 degree and 90 degrees in relation to the headrest (Fig 2 Panels A and B).

7. You report “The Petri dishes were kept open for 15 minutes” – was this 15 mins after the 1 min procedure (i.e. 16 mins total) or including the 1 min procedure (i.e. 15 mins total). How was a 15-minute exposure time decided upon?

A: We have rephrased the sentence to clarify this. Please see the exert below from the revised manuscript.

“The Petri dishes were open immediately before activating the handpiece and were kept open for 15 minutes thereafter. They were then placed in an aerobic incubator (Tecnal, TE-399, Piracicaba, Brazil) for 48 hours.”

8. Were any negative control conditions used, i.e. Petri dishes exposed without a procedure performed?

A: Thank you for your careful reading. We have included information to the methods clarify this. Please see the exert below from the revised manuscript.

“As negative controls, Petri dishes enriched medium to Lactobacillus spp. were left open for 15 minutes prior to the test.”

9. The legend for Figure 1 should come after the references. This legend reports: “The image filter used for the accompanying figures at the bottom was the Qscan (AioBio, Seoul, South Korea).” but this technique is not reported in the methods – it appears to be a form of direct fluorescence technique.

A: Thank you for your careful reading, we have included additional information to the methods, as follows:

“With the handpieces activated using fluorescence Qscan (AioBio, Seoul, South Korea), images from droplets and aerosols were recorded (Fig 2 Panels F - H).”

10. Did you conduct an a priori power calculation? If so, report this. ANOVA was used to test significance between distances, what test was used between experiments (i.e DD/US/US+Device)?

A: The number of petri dishes and the number of repetitions were based on a previous preliminary study (Montalli et al., 2020, reference 8). This was further reinforced by the relatively low standard deviation values observed in all tested groups.

Results

1. You report that there was a reduction in CFU when using the US compared to the air-turbine; this is not a reduction – it is a difference. One would not use an US in place of an air-turbine (except in a select number of cases) and it is clearly a very different experimental condition to examine. Use difference instead of reduction throughout the manuscript.

A: Thank you for your careful reading. We have modified the text to accommodate this interesting observation. 

“Regarding CFU counts and bioaerosol dispersion (when distance from the source was taken into account), a significant difference was observed between the dental drill (DD) and the ultrasonic tip (UT), as well as the ultrasonic tip combined with the spray control device (UT-SC).”

2. Table 1: n is missing for the US-SC condition in the heading. No significance testing is reported – or at all elsewhere in the results for that matter. These need to be reported somewhere. Reporting of this data may be aided by also using a bar chart.

A: Thank you for your careful reading, we have changed the table into a bar chart to facilitate visualization. Additionally, we changed the word reduction to “lower CFU count” with an asterisk * to indicate a significant difference.

3. Table 2: again, no significance testing reported. It would be more appropriate to report the procedure with the lower CFU as a proportion of the procedure with the larger CFU, because as mentioned above you are not really showing a reduction, only a difference. For example, for DD versus US, it may be better to say that mean CFU for US was (x)% of that for DD rather than reporting a 95.57% reduction.

A: Thank you for your careful reading. We have excluded Table 2 and the mean CFU count was reported in the results of the new version of the manuscript.

“Regarding the dental drill (DD) at 90º and 0º angles, higher CFU counts were observed when compared with the ultrasonic tip (UT). At 50 cm, the mean and standard deviation (SD) of CFU of Lactobacillus casei Shirota for DD was 13554.60 (4071.03) CFU, while for UT was 286.67 (73.99) CFU (p <0.0001). At 100 cm, the mean DD was 7761.64 (2073.74) CFU and UT was 290.84 (96.21) CFU (p <0.0001). The mean (SD) at 150 cm for DD was 4464.00 (1011.65) CFU, while for UT was 344.33 (36.19) CFU (p <0.0001) (Fig 4). The mean reduction between DD and UT was 96.42%. When using the spray control device on the ultrasound tip (UT-SC), the mean CFU counts (standard deviation) at 50 cm, 100 cm and 150 cm were 4.50 (0.58) CFU, 3.67 (1.06) CFU and 4.17 (3.47 ) CFU, which compared to the UT data translated into a significant difference (p <0.0001). The average reduction between UT and UT-SC was 98.66%. When comparing DD and UT-SC, the mean reduction was 99.95%.”

Discussion

1. In the first paragraph, you mention proportions of health care workers among confirmed COVID-19 cases in Italy and Spain – when was this? The time point is relevant in an evolving pandemic. Were these point prevalences at this particular time point or cumulative proportions of health workers who had been infected at any time?

A: Thank you for your careful reading. We have included the month of the report used as a reference.

2. In paragraph 2 you cite “The National Federation of Medical Doctors and Dentists”; which nation is this? You suggest that the data in this paragraph suggest that infection among oral health professionals is due to occupational exposure, but they do not unless you also know the corresponding rate of infection in the general population at that time to compare it to. You do not report this.

A: We have included the nation (Italy) which this reference is related to, as well as the total number of confirmed cases in the date of the number reported in the sentence.

“Data from The National Federation of Medical Doctors and Dentists (Italy) updated on March 28th report the death of 86 medical doctors and 8 dentists over 92.472 confirmed cases [23]. This reinforces the concept that close contact with positive patients, whether symptomatic or not, exposes health workers to a higher risk of infection [25].”

3. In line 209 you mention “record[ing] the temperature of each team and patient as a routine procedure”; what is the evidence for the effectiveness of this?

A: There is no evidence of such effectiveness, however, this has been adopted by several local authorities as “best practice” and endorsed by medical associations. Since this information was illustrative and not critical to the manuscript, we have removed it from the text.

4. In line 227 you state that bacterial models of dental bioaerosol “thus [provide] more reliable results on dispersion of bioaerosols generated during the experiments than germ-free models”. Where is your evidence for this compared to other methodologies such as tracer dyes or particle counters? In fact, you need to discuss the evidence from such non-biological studies on dental aerosols and consider your findings in relation to these. A number of reviews of the literature in this area have recently been published and may be helpful in your locating of such primary sources of evidence: (https://www.scottishdental.org/wp-content/uploads/2020/08/Ventillation-Final-Copy-1.pdf ; https://www.sdcep.org.uk/published-guidance/covid-19-practice-recovery/rapid-review-of-agps/ ; https://www.medrxiv.org/content/10.1101/2020.08.28.20183475v1)

A: Thank you for your suggestion. We have rephrased the discussion in order to accommodate the important points raised by the referee herein. Please see the exert below from the revised version of the manuscript.

“A recent systematic review investigated contamination associated with routine dental procedures and concluded that ultrasonic scaling was one of the highest in contamination levels [29].”

29. Innes N, Johnson LG, Al-Yaseen W, Harris R, Jones SKC, McGregor S, Robertson M, Wade WG, Gallagher JE. A systematic review of droplet and aerosol generation in dentistry. medRxiv preprint doi: https://doi.org/10.1101/2020.08.28.20183475

Limitations

You discuss the limitations of your study, however you have failed to identify the most significant limitations, as follows, which must be considered and reported:

1. You report that you have detected “aerosol” and have concluded that US produces less than the air-turbine. You have not identified the fact that your method will mainly detect droplet contamination, and the proportion of aerosols that settle onto Petri dishes in 15 minutes. You have detected neither aerosol which settles after 15 minutes, nor that which remains suspended in the air. This suspended aerosol is likely to be the greatest inhalation risk during a dental aerosol generating procedure. You should explore this limitation in the discussion and its implications to your findings.

2. The procedure you used is not equivalent to a standard dental procedure in terms of procedure or time. This will affect the ability to translate to actual dental procedures.

3. You examined to 1.5 m from the procedure, what happens beyond this? This limitation has relevance for open-plan dental clinics such as those where the investigation was performed.

4. You equate your findings with a bacterial biomarker to the carriage of virus (SARS-CoV-2). You need to discuss the limitation that a virion (which is much smaller, may be charged etc.) may not behave in the same way as a bacterium, and this will affect the translatability of your findings. The requirement for bacteria to be viable for you to detect it is also a limitation, and this may be different to the infectivity of virions at any given distance.

5. You do not discuss that placing the bacterial biomarker into dental irrigation systems will tell you where water spray will be distributed to, but not necessarily where potential pathogens from the mouth will be picked up by this water spray and distributed to. This affects how relevant your findings are to the risk of transmission from dental procedures in real-world conditions.

6. Finally, your findings show that US likely produces less contamination than an air-turbine, and that the spray control device reduces US contamination, but we still do not know what the risk threshold of SARS-CoV-2 is. For example, what viral load is required to cause COVID-19? Will the reduced amount of contamination from US with the spray device in an infected individual still be significant transmission risk? We do not yet know this and this should be discussed.

A: Thank you for all limitations raised. The authors agree that all are important, and they have been included in the revised manuscript. Please see the revised discussion chapter.

---

## [Decision Letter · Decision Letter 1]

15 Dec 2020

PONE-D-20-28707R1

A novel dental biosafety device to control the dispersion of bioaerosols from dental ultrasonic tips.

PLOS ONE

Dear Dr. Montalli,

Thank you for submitting your manuscript to PLOS ONE. After careful consideration, we feel that it has merit but does not fully meet PLOS ONE’s publication criteria as it currently stands. Therefore, we invite you to submit a revised version of the manuscript that addresses the points raised during the review process.

We look forward to receiving your revised manuscript.

Kind regards,

Ratilal Lalloo

Academic Editor

PLOS ONE

Additional Editor Comments (if provided):

Dear authors;

Two reviewers continue to question the use of the term 'bioaerosols' and also have a few additional queries and concerns.

Reviewers' comments:

**Comments to the Author**

1. If the authors have adequately addressed your comments raised in a previous round of review and you feel that this manuscript is now acceptable for publication, you may indicate that here to bypass the “Comments to the Author” section, enter your conflict of interest statement in the “Confidential to Editor” section, and submit your "Accept" recommendation.

Reviewer #1: All comments have been addressed

Reviewer #2: All comments have been addressed

Reviewer #3: (No Response)

2. Is the manuscript technically sound, and do the data support the conclusions?

Reviewer #1: Yes

Reviewer #2: Partly

Reviewer #3: Yes

3. Has the statistical analysis been performed appropriately and rigorously? 

Reviewer #1: Yes

Reviewer #2: N/A

Reviewer #3: Yes

4. Have the authors made all data underlying the findings in their manuscript fully available?

Reviewer #1: Yes

Reviewer #2: Yes

Reviewer #3: Yes

5. Is the manuscript presented in an intelligible fashion and written in standard English?

Reviewer #1: Yes

Reviewer #2: Yes

Reviewer #3: Yes

6. Review Comments to the Author

Reviewer #1: The revisions have greatly improved the paper and my concerns have been addressed.

I noted one error in the new text - line 162 -

"As negative controls, Petri dishes enriched medium to Lactobacillus spp. were left open for 15 minutes..."

There are some missing words in this sentence.

Reviewer #2: The authors did not fully address the comments. The main scientific error was the claim of "bioaerosols" in the title and throughout the manuscript. The authors should provide the evidence of aerosols as the aerosols are <5um, while the size of Lactobacillus casei is ranging 0.7-1.1 x 2.0-4.0 micrometer. How can the authors know the dental drill can generate aerosols with Lactobacillus casei. How do you know they are not droplets instead? This is a serious scientific error which is unacceptable.

Regarding the S. Fig 1, In the groups of DD 50, 100 and 150 cm, the colonies were too many to count (particularly the DD 50cm group). How did the authors can distinguish the boundaries between two overlapped colonies. How many replicates and how many experiments have been performed to avoid the bias?

The authors used COVID as background, however, the bacteria was used for the experiment. The size of SARS-CoV2 is ~100nm and the bacterial is bigger than 2um; how did the authors correlate their findings with SARS-CoV2 transmission route.

Reviewer #3: Thank you for revising the manuscript. The changes certainly improve the work. I still have some minor concerns however:

1. Using the term bioaerosols in the title and throughout the manuscript is misleading, as you have measured splatter, droplets, and the proportion of aerosols which have settles onto surfaces. You do not know which of these components make up the contamination you have detected, nor have you measured suspended aerosols. This needs to be brought out more in the discussion (although I acknowledge you have mentioned it, albeit briefly). I would also suggest changing the title to “A novel dental biosafety device to control the dispersion of contamination from dental ultrasonic tips” or similar to reflect this and correcting the manuscript throughout.

2. On line 175, you report the petri dishes as 90 cm diameter. I assume you mean 90 mm.

3. The manuscript continues to describe a reduction between DD and UT (lines 207, 212, 272). As I have mentioned this is a difference not a reduction and this should be corrected.

4. Your experiments were conducted in an unventilated setting which allows you to control for the effects of this, however you should make this very clear in your discussion, and emphasise that the CFU observed are likely to be lower in settings with mechanical ventilation.

5. When reporting significance tests, please report the test statistic, degrees of freedom, and significance values. Useful guidance on this can be found here: http://evc-cit.info/psych018/Reporting_Statistics.pdf

and here: http://courses.washington.edu/ll317/psych318/files/APAstats.pdf

7. Reviewer #1: **Yes: **Laurence J. Walsh

Reviewer #2: No

Reviewer #3: **Yes: **James R Allison

---

## [Author Response · Author response to Decision Letter 1]

28 Jan 2021

We would like to thank you the reviewers for all constructive comments, and we are certain that they were key to improving the manuscript. We accepted all the suggestions and added information to the text to clarify the points raised by the referees. All modifications were highlighted in red. Our responses to the individual comments are listed below. We hope that these corrections will satisfy the reviewer’s comments.

Sincerely yours,

Reviewer #1: 

The revisions have greatly improved the paper and my concerns have been addressed.

I noted one error in the new text - line 162 -

"As negative controls, Petri dishes enriched medium to Lactobacillus spp. were left open for 15 minutes..."

There are some missing words in this sentence.

A: Thank you for your careful reading. We have rephrased the error in the new text as requested. Below the new information included in the revised manuscript.

“As negative controls, Petri dishes containing MRS medium were left open for 15 minutes prior to the test.”

Reviewer #2: 

The authors did not fully address the comments. The main scientific error was the claim of "bioaerosols" in the title and throughout the manuscript. The authors should provide the evidence of aerosols as the aerosols are <5um, while the size of Lactobacillus casei is ranging 0.7-1.1 x 2.0-4.0 micrometer. How can the authors know the dental drill can generate aerosols with Lactobacillus casei. How do you know they are not droplets instead? This is a serious scientific error which is unacceptable.

A: Thank you. The authors have modified the terms aerosol and droplets to PCDP (potentially contaminated dispersion particles), as they understand that such term encompasses most types of airborne particles that could potentially carry pathogens. Should the reviewer find this term inappropriate the authors would be most grateful for a suggestion on a suitable replacement.

Regarding the S. Fig 1, In the groups of DD 50, 100 and 150 cm, the colonies were too many to count (particularly the DD 50cm group). How did the authors can distinguish the boundaries between two overlapped colonies. How many replicates and how many experiments have been performed to avoid the bias?

A: According to the standard operating procedures (SOP) followed by the technicians that performed the CFU counting an L. casei Shirota colony measures on average 2mm (SD 1mm). Whenever colonies were overlapped, the longest length of the joined colonies (mm) was divided by 2 (mm) thus establishing the approximate number of overlapping colonies.

The authors used COVID as background, however, the bacteria was used for the experiment. The size of SARS-CoV2 is ~100nm and the bacterial is bigger than 2um; how did the authors correlate their findings with SARS-CoV2 transmission route.

A: The experimental model of particle dispersion presented herein aimed at investigating the spread of droplets generated during common dental procedures. The focus were the particles. Such particles (droplets) are generally larger than both viruses and bacteria and therefore have the potential to harbour and carry either or both (even many of each). In the current context of pandemic, therefore, the L. casei Shirota was merely used as a biologically safe surrogate marker of SARS-CoV-2.

Reviewer #3: Thank you for revising the manuscript. The changes certainly improve the work. I still have some minor concerns however:

1. Using the term bioaerosols in the title and throughout the manuscript is misleading, as you have measured splatter, droplets, and the proportion of aerosols which have settles onto surfaces. You do not know which of these components make up the contamination you have detected, nor have you measured suspended aerosols. This needs to be brought out more in the discussion (although I acknowledge you have mentioned it, albeit briefly). I would also suggest changing the title to “A novel dental biosafety device to control the dispersion of contamination from dental ultrasonic tips” or similar to reflect this and correcting the manuscript throughout.

A: Thank you for your suggestion. The authors accepted the new title suggested considering also the observations by Reviewer 2 regarding the definition of the particles.

2. On line 175, you report the petri dishes as 90 cm diameter. I assume you mean 90 mm.

A: Thank you for spotting this error. We have made the appropriated correction as noted. 

3. The manuscript continues to describe a reduction between DD and UT (lines 207, 212, 272). As I have mentioned this is a difference not a reduction and this should be corrected.

A: Thanks for this observation. The text has been corrected as suggested. 

4. Your experiments were conducted in an unventilated setting which allows you to control for the effects of this, however you should make this very clear in your discussion, and emphasise that the CFU observed are likely to be lower in settings with mechanical ventilation.

A: The authors accepted this suggestion and have added this information to the discussion as pasted below: 

“It is important to highlight, however, that different dental procedures and settings may affect the dynamics of droplets and aerosol [29], for instance the fact that the experiment presented herein were performed with no mechanical ventilation.”

5. When reporting significance tests, please report the test statistic, degrees of freedom, and significance values. Useful guidance on this can be found here: http://evc-cit.info/psych018/Reporting_Statistics.pdf

and here: http://courses.washington.edu/ll317/psych318/files/APAstats.pdf

A: Thank you for your careful reading, we have included additional description to the results as pasted below:

“Regarding CFU counts and PCDP dispersion (when distance from the source was taken into account), an analysis of variance showed a significant difference was observed between the dental drill (DD) and the ultrasonic tip (UT), F(5.30) = 703.90, p = <0.0001, as well as the ultrasonic tip combined with the spray control device (UT-SC), F(5.30) = 62.76, p = <0.0001. The representative images of the Petri dishes of each group are found as supplementary material (Supplemental 1). 

Regarding the dental drill (DD) at 90º and 0º angles, higher CFU counts were observed when compared with the ultrasonic tip (UT). Post hoc analyses using the Tukey’s multiple comparisons criterion for significance indicated that at 50 cm, the mean and standard deviation (SD) of CFU of Lactobacillus casei Shirota for DD was 13554.60 (4071.03) CFU, while for UT was 286.67 (73.99) CFU (p <0.0001). At 100 cm, the mean DD was 7761.64 (2073.74) CFU and UT was 290.84 (96.21) CFU (p <0.0001). The mean (SD) at 150 cm for DD was 4464.00 (1011.65) CFU, while for UT was 344.33 (36.19) CFU (p <0.0001) (Fig 4). The mean difference between DD and UT was 96.42%. When using the spray control device on the ultrasound tip (UT-SC), the mean CFU counts (standard deviation) at 50 cm, 100 cm and 150 cm were 4.50 (0.58) CFU, 3.67 (1.06) CFU and 4.17 (3.47 ) CFU, which compared to the UT data translated into a significant difference (p <0.0001). The average difference between UT and UT-SC was 98.66%. When comparing DD and UT-SC, the mean difference was 99.95% (Fig. 4).”

---

## [Decision Letter · Decision Letter 2]

1 Feb 2021

A novel dental biosafety device to control the spread of potentially contaminated dispersion particles from dental ultrasonic tips.

PONE-D-20-28707R2

Dear Dr. Montalli,

We’re pleased to inform you that your manuscript has been judged scientifically suitable for publication and will be formally accepted for publication once it meets all outstanding technical requirements.

Kind regards,

Ratilal Lalloo

Academic Editor

PLOS ONE

Additional Editor Comments (optional):

Reviewers' comments:

Reviewer's Responses to Questions

**Comments to the Author**

Reviewer #2: All comments have been addressed

2. Is the manuscript technically sound, and do the data support the conclusions?

Reviewer #2: Yes

3. Has the statistical analysis been performed appropriately and rigorously? 

Reviewer #2: Yes

4. Have the authors made all data underlying the findings in their manuscript fully available?

Reviewer #2: Yes

5. Is the manuscript presented in an intelligible fashion and written in standard English?

Reviewer #2: Yes

6. Review Comments to the Author

Reviewer #2: The authors have addressed all the comments. There is no further comments required before the publication.

7. **Do you want your identity to be public for this peer review?**

Reviewer #2: No

---

## [Editor Report · Acceptance letter]

3 Feb 2021

PONE-D-20-28707R2 

A novel dental biosafety device to control the spread of potentially contaminated dispersion particles from dental ultrasonic tips. 

Dear Dr. Montalli:

I'm pleased to inform you that your manuscript has been deemed suitable for publication in PLOS ONE. Congratulations! Your manuscript is now with our production department. 

Kind regards, 

on behalf of

Dr. Ratilal Lalloo 

Academic Editor

PLOS ONE